# Characteristics and Application of a Novel Cold-Adapted and Salt-Tolerant Protease EK4-1 Produced by an Arctic Bacterium *Mesonia algae* K4-1

**DOI:** 10.3390/ijms24097985

**Published:** 2023-04-28

**Authors:** Hailian Rao, Ran Huan, Yidan Chen, Xun Xiao, Wenzhao Li, Hailun He

**Affiliations:** 1School of Life Sciences, Central South University, Changsha 410013, China; 2Key Laboratory of Functional Dairy, Co-constructed by Ministry of Education and Beijing Municipality, College of Food Science and Nutritional Engineering, China Agricultural University, Beijing 100000, China

**Keywords:** *Mesonia algae* K4-1, M4 peptidase family, cold-adapted, substrate specificity, antioxidant

## Abstract

*Mesonia algae* K4-1 from the Arctic secretes a novel cold-adapted and salt-tolerant protease EK4-1. It has the highest sequence similarity with Stearolysin, an M4 family protease from *Geobacillus stearothermophilus*, with only 45% sequence identity, and is a novel M4 family protease. Ek4-1 has a low optimal catalytic temperature (40 °C) and is stable at low temperatures. Moreover, EK4-1 is still active in 4 mol/L NaCl solution and is tolerant to surfactants, oxidizing agents and organic solvents; furthermore, it prefers the hydrolysis of peptide bonds at the P1’ position as the hydrophobic residues, such as Leu, Phe and Val, and amino acids with a long side chain, such as Phe and Tyr. Mn^2+^and Mg^2+^ significantly promoted enzyme activity, while Fe^3+^, Co^+^, Zn^2+^ and Cu^2+^ significantly inhibited enzyme activity. Amino acid composition analysis showed that EK4-1 had more small-side-chain amino acids and fewer large-side-chain amino acids. Compared with a thermophilic protease Stearolysin, the cold-adapted protease EK4-1 contains more random coils (48.07%) and a larger active pocket (727.42 Å^3^). In addition, the acidic amino acid content of protease EK4-1 was higher than that of the basic amino acid, which might be related to the salt tolerance of protease. Compared with the homologous proteases EB62 and E423, the cold-adapted protease EK4-1 was more efficient in the proteolysis of grass carp skin, salmon skin and casein at a low temperature, and produced a large number of antioxidant peptides, with DPPH, ·OH and ROO· scavenging activities. Therefore, cold-adapted and salt-tolerant protease EK4-1 offers wide application prospects in the cosmetic and detergent industries.

## 1. Introduction

Proteases are a class of enzymes that hydrolyze protein peptide chains and can be divided into different groups according to their active site and catalytic mechanism [1,2]. To date, there are 273 different families of protein hydrolases in the MEROPS database. The vast majority of these proteins are derived from microorganisms [3]. Microbial proteases not only play an important role in cellular metabolic processes, but are also important sources of industrial proteases. They account for about 2/3 of total enzyme sales worldwide [4], and are highly favored in detergent, food processing, feed production, peptide synthesis, leather processing and biodegradation industries. However, there are still some limitations in the industrial application of commercial proteases, such as low enzyme activity at low temperature, poor salt tolerance, low tolerance to chemical reagents, single cleavage site, high price and so on. Therefore, it remains a challenge to find and develop novel proteases that tolerate one or more extreme conditions.

Cold-adapted proteases present a class of enzymes that can be catalyzed at a low temperature (the optimum temperature is <40 °C) with high efficiency, and are usually still active at 0 °C [5]. Most cold-adapted proteases are produced by cold-adapted microorganisms. Cold-adapted microorganisms can adapt to low-temperature environments, their optimal growth temperature is around 15 °C and they can survive in 0 °C or even −20 °C [6]. These microorganisms generally live in the deep sea and the Arctic and South Poles, as well as refrigerators, cold storage and other artificial low-temperature environments [7]. In order to adapt to the cold living environment, the extracellular proteases produced by the cold-adapted microorganisms also have typical cold-resistance characteristics, such as higher catalytic activity at low temperatures, lower optimal catalytic temperature and higher thermal sensitivity [8]. The application of cold-adapted microorganisms and their secreted extracellular cold-adapted proteases has received increasing attention in the last three decades. Hao et al. found that the cold-adapted protease purified from *Pseudomonas aeruginosa* HY1215 showed valuable stability against commercially available surfactants and bleaching agents, which could be evaluated as a cold-washing detergent enzyme [9]. The study of He et al. showed that cold-adapted protease MCP-01 produced by the deep-sea bacterium *Pseudoaltermonas* sp. SM9913 could improve the flavor of frozen meat better than mesophilic proteases by providing additional taste and releasing essential amino acids [10]. Cold-adapted protease had incomparable advantages over mesophilic protease in food, cosmetic, waste treatment, pharmaceutical and other industries, and had good application prospects; thus, it is necessary to explore and develop new cold-adapted proteases.

*Mesonia algae* K4-1 is a moderately halophilic and psychrophilic bacterium that was isolated from the Arctic Ocean in our laboratory [11]. In order to adapt to the special polar marine environment, *Mesonia algae* K4-1 synthesized a large number of extracellular polysaccharides and secreted different families of cold-adapted extracellular proteases [11]. The genus *Mesonia* was created by Nedashkovskaya et al. with *Mesonia algae* as the type species and belonged to the family Flavobacterium of the phylum Bacteroidetes. The genus *Mesonia* currently consists of ten species with validly published names, as of July 2022 (https://lpsn.dsmz.de/genus/mesonia (accessed on 10 March 2023)) [12]. In this study, the major extracellular protease EK4-1 secreted by *Mesonia algae* K4-1 was isolated and purified, and its cold adaptation and salt tolerance properties were analyzed based on the amino acid composition and three-dimensional structure characteristics. Then, the enzymatic kinetic parameters and cleavage sites were tested using the B chain of insulin and synthetic dipeptides as the substrates. In addition, a comparison of the degradation efficiency of EK4-1 with other homologous proteases at different temperatures for low-value protein substrates demonstrated the superiority of the rate of the enzymatic digestion of EK4-1 at low temperatures.

## 2. Results

### 2.1. Purification of Protease EK4-1

After fermentation, the highest enzyme activity in the *Mesonia algae* K4-1 fermentation broth was observed at 96 h. The protease, named EK4-1, was purified via (NH_4_)_2_SO_4_ gradient precipitation, DEAE anion-exchange chromatography and size-exclusion chromatography using a Superdex-75 column and a chromatographic FPLC system (Bio-Rad, Hercules, CA, USA). Under optimum assay conditions, the purified enzyme had a specific activity of 893 U•mg^−1^, with a final yield of 7.53% and purification fold of 14 fold (Table 1). The molecular weight of protease EK4-1 was estimated to be 37 kDa based on SDS-PAGE (Appendix A).

### 2.2. Enzymatic Properties of EK4-1

The optimum temperature for protease EK4-1 was determined by measuring the efficiency of EK4-1 hydrolysis casein at different temperatures (Figure 1A). The results showed that the optimal reaction temperature of EK4-1 for casein hydrolysis was about 40 °C, and the enzyme activity could be maintained at 22% at 4 °C. The thermal stability of protease EK4-1 showed that EK4-1 was stable at 30 °C, and the enzyme activity remained at about 93% after 120 min (Figure 1B). However, the thermal stability of the enzyme at 40 °C was poor, with a half-life of about 35 min. At 50 °C for 5 min, the enzyme activity lost nearly 92%. To assess the thermotropic behavior of protease EK4-1, differential scanning calorimetry (DSC) was used as it allows for the detection of conformational changes and the structural stability of proteins. Protease EK4-1 showed a heat peak at 61.52 °C (Figure 1C). Compared with E495 of the M4 family secreted by Arctic Sea ice bacteria *Pseudoalteromonas* sp.SM495, the optimum enzyme activity temperature was 57 °C, and the enzyme activity at 0 °C was only 1.2% of the highest enzyme activity. The optimal enzyme activity temperature of EK4-1 was much lower, and the cold-adapted protease EK4-1 had a better effect on performing a catalytic function at a low temperature. These results indicated that EK4-1 had higher activity at a low temperature [8,13,14]. As show in Figure 1D, EK4-1 had high catalytic activity under neutral conditions, and the optimal pH was 8.0. After mixing EK4-1 in different pH buffer solutions for 4 h, EK4-1 could still maintain more than 70% enzyme activity in the pH range of 7.0–9.0 (Figure 1E).

The effect of NaCl on the catalytic activity of protease EK4-1 showed that the activity of EK4-1 increased first and then decreased with an increase in salt concentration. When the concentration of NaCl was 0.5 M, the activity of protease was the best, which was almost twice as high as that without salt, and even at 4.5 M NaCl concentration, 60% of the enzyme activity was still maintained. The results of high-salt-tolerance experiments showed that the relative enzyme activity was 110% after 48 h incubation with 1.0 M NaCl, and the enzyme activity remained at 58% at a salt concentration of 4.0 M NaCl (Figure 1H). The salt activation and salt tolerance of EK4-1 indicated the potential application of EK4-1 in high-salt environments, such as soy sauce production and fish sauce fermentation. Many studies have found that in addition to having high catalytic activity under a high salt concentration, salt-tolerant enzymes were likely to be stable in the same organic solvent medium with low available water activity, which was a valuable tool for biocatalysts in water–organic media [15]. EK4-1 could maintain more than 50% activity in isopropanol, DMSO, xylene, n-propanol and hexane, while ethanol, methanol and isoamyl alcohol had strong inhibitory effects on enzyme activity (Figure 1I). The effects of oxidant H_2_O_2_, non-ionic surfactants Tween 80 and Triton x-100 and anionic surfactant SDS on the enzyme activity of protease EK4-1 were further studied (Figure 1F). The enzyme can maintain more than 50% activity under the conditions of 1% H_2_O_2_, 1% Tween 80 and 1% tritonx-100, compatible with detergent formulation. Protease EK4-1 could be widely used in the detergent industry due to its good properties of being cold-adapted, salt-tolerant and detergent-resistant. Among the metal ions tested, 10 mM Mn^2+^ and Mg^2+^ can enhance the activity of EK4-1 to 132.3% and 146.6%, respectively. EK4-1 was relatively stable in 2.5 mM and 10 mM K^+^, Ca^2+^ and Ba^2+^, and its activity was more than 75%, while the catalytic activity of EK4-1 could be significantly inhibited by Fe^3+^, Co^2+^, Zn^2+^ and Cu^2+^ (Figure 1J).

### 2.3. Structural Properties of EK4-1

Purified protease EK4-1 was identified using MALDI-TOF/TOF mass spectrometry and then the obtained sequence fragments were searched for in *Mesonia algae* K4-1 genome data (GenBank: GCA_008017825.1) [11]. The gene-encoding EK4-1 was 2955 bp in length, and it was assigned the accession number WP_147863623.1. The analysis of the InterPro database (https://www.ebi.ac.uk/interpro/ (accessed on 10 March 2023)) estimated that EK4-1 consisted of several domains, including the FTP domain (InterPro: IPR011096), Peptidase M4 domain (InterPro: IPR013856), Peptidase M4 C-terminal domain (InterPro: IPR001570), FN3 domain (InterPro: IPR001570), GEVED domain (InterPro: IPR045474) and Secre_tail domain (InterPro: IPR26444) [16] (Figure 2A). After conducting similarity analysis with various identified homologous proteases in the MEROPS database, it was found that protease EK4-1 had the highest sequence similarity with the identified M4 family Stearolysin from *Geobacillus stearothermophilus* [17], with a sequence identity of 45%. This result predicted that EK4-1 may be a novel M4 family protease (Figure 2B) and the activity of protease EK4-1 was inhibited by the metalloprotease inhibitor 1,10-phenanthroline monohydrate (OP) (Appendix A).

To date, most bacterial extracellular metalloproteases found are endopeptidases containing one zinc ion in the active center, which forms four coordination bonds with three amino acid residues and a water molecule to construct the active center of a tetrahedral structure, usually two His and one Glu [18]. The EK4-1 gene encodes a precursor of 984 residues. The BLAST search of the MEROPS databases using the EK4-1 precursor sequence as a query revealed several sequences with high identities. The metalloprotease Stearolysin identified from *Geobacillus stearothermophilus* showed the highest similarity (45%) to the EK4-1 precursor. The neutral peptidase identified from *Thermoactinomyces* sp. 27A showed the second highest similarity (44.3%) to the EK4-1 precursor. Thermolysin identified from the strain *Bacillus thermoproteolyticus* showed the third highest similarity (44%). The alignment of mature sequences of EK4-1, Stearolysin, neutral peptidase and thermolysin is shown in Figure 2B, and they all had the conserved HEXXH sequence of typical metalloproteases. In contrast to the other three proteases, the third and fourth amino acid in the conserved region of protease EK4-1 are Ile and Gly, respectively (Figure 2B).

Studies have reported that cold-adapted proteases generally have high structural flexibility and poor thermal stability [19]. Xie et al. studied MCP-02 from a deep-sea bacterium, E495 from Arctic Sea ice bacterium and homolog mesophilic metalloprotease pseudolysin from a terrestrial bacterium, which showed that the flexibilities of the proteins were pseudolysin < MCP-02 < E495, suggesting that increased flexibility was a strategy for cold adaptation [20]. Compared with E495, MCP-02 and Stearolysin, EK4-1 had more small-side-chain amino acids, such as Gly, Ser, Thr and Asn, and fewer large-side-chain amino acids, such as Arg, Leu and Lys (Table 2). This result was consistent with the amino acid composition characteristics of cold-adapted proteins that had been reported [21]. It is worth mentioning that the cold-adapted characteristics of EK4-1 were more obvious than those of E495. In addition to the differences in amino acid composition, cold-adapted proteases usually contained higher amounts of random coils to increase the flexibility [22]. The prediction of the secondary structure of the metalloproteases showed that the contents of the random coil in EK4-1, E495, MCP-02 and Stearolysin decreased in the order of 48.07%, 47.12%, 43.47% and 42.88%, respectively. The large number of random coils in EK4-1 helped it to be more flexible in space structure, meaning it was easier to unfold and produced conformational changes. In addition to cold adaptation, salt tolerance was another distinctive characteristic of EK4-1. Moshe’s research showed that acidic amino acids made the protein surface have a large number of negative charges, so that the protease could maintain a certain flexibility in the high-salt environment to ensure the catalytic activity of the enzyme [23]. On the other hand, the acidic side chain can bind water and salt ions, which makes it possible for proteins to adapt to the high-salt environment and maintain stability, and also increases the solubility of proteins in the high-salt environment [24]. Compared with the other three homogenous metalloproteases, the amino acid composition of EK4-1 from Arctic Sea water showed that the amount of acidic amino acids was much higher than that of basic amino acids, which was suitable for the sequence characteristics of salt-tolerant proteases (Table 2).

Cold-adapted protease EK4-1 and thermophilic protease Stearolysin were modeled via the Swiss-Model website and then the active pocket size of the protease catalytic domain was calculated using Proteins Plus. The calculated results showed that the catalytic pocket volume of cold-adapted protease EK4-1 was 727.42 Å^3^ (Figure 3A) and that of thermophilic protease Stearolysin was 538.43 Å^3^ (Figure 3B). The active pocket of cold-adapted protease EK4-1 was much larger than that of Stearolysin. The larger catalytic cavity might allow the cold-adapted protease EK4-1 easier binding of substrates. This might also be an adaptation of the protease EK4-1 to the low-temperature environment.

To analyze the cleavage sites of protease EK4-1, digestion was performed using insulin B chain as the substrate. The peptide fragments obtained after hydrolysis were then analyzed using HPLC and mass spectrometry (Figure 4A). The molecular weight of 14 peptides with a high abundance was determined, the amino acid composition of these 14 peptides was matched in the FindPept database (Table 3) and cleavage sites were further analyzed (Figure 4B,C). The insulin B chain was hydrolyzed to produce multiple cleavage sites, which were, respectively, His + Leu, Leu + Cys, Gly + Ser, Glu + Ala, Ala + Leu, Leu + Tyr, Tyr + Leu, Leu + Val, Arg + Gly and Gly + Phe. These results indicated that most of the P1’ positions were hydrophobic amino acids such as Leu, Ala and Val, and amino acids with a large side chain, such as Phe and Tyr. The substrate specificity of EK4-1 is similar to other proteases of the M4 family, such as aureolysin and thermolysin, and the specific hydrolyzed P1’ position is a large hydrophobic amino acid [25]. Compared with the M4 family protease MCP-02, four cleavage sites of EK4-1 were the same as those of MCP-02 (Leu + Cys, Ala + Leu, Leu + Tyr and Arg + Gly). In addition, protease EK4-1 also had a certain preference for hydrophobic amino acids and long-side-chain amino acids at the P1 position. However, MCP-02 had a weak preference for these amino acids. In addition, protease EK4-1 displayed more cleavage sites for the B chain of insulin. More cleavage sites could help *Mesonia algae* K4-1 to better degrade extracellular protein particles for nutrients, which might be an adaptation of Arctic bacteria to extreme environments.

To further confirm the digestion characteristics of EK4-1, six peptides were synthesized and hydrolyzed via protease EK4-1. EK4-1 could degrade the dipeptide substrates FA-Gly-Leu-NH_2_ and FA-Gly-Phe-NH_2_ with large side chains and strong hydrophobicity at the P1’ position. The hydrolysis rate of FA-Gly-Phe-NH_2_ with the aromatic amino acid Phe at the P1’ position was the fastest, which was 1.20 times higher than that of FA-Gly-Leu-NH_2_ as the hydrolysis substrate (Table 4, Figure 5). However, unlike the protease MCP-02, EK4-1 could not degrade FA-Gly-Val-NH2, suggesting that it might also select amino acids at the P position of the substrate.

### 2.4. EK4-1 Efficiently Degrades Low-Value Protein Substrates at Low Temperature

The aquatic product processing industry leads to the production of a large number of waste products (skin, bone, head, viscera and scales). However, in addition to being used as fertilizer or feed, most of these low-value fish products and their by-products are treated as garbage, resulting in a huge waste of resources and environmental pollution, and are a source of low-value protein. Recent studies have shown that the hydrolysis of these protein-rich fish by-products with proteases can produce a variety of bioactive peptides which can be used as additives in functional foods and have high economic value that enhance the added value of the protein.

Our results showed that the cold-adapted protease EK4-1 has high catalytic efficiency at a low temperature and multiple cleavage sites. In order to further explore the possibility and superiority of cold-adapted protease EK4-1 in industrial applications, we used protease EK4-1 to proteolyze three protein substrates (grass carp skin, salmon skin and casein) at different temperatures and compared the catalytic efficiency of the homologous metalloproteases E423 and EB62. The results showed that the content of -NH_2_ in the reaction system increased significantly after protease digestion, and the content of free -NH_2_ could be used to indicate the degree of hydrolysis of the substrate protein. The hydrolysis degree (DH) of different substrates treated with the three proteases at 40 °C was defined as being 100%. The DH of three enzymes EK4-1, E423 and EB62 for different substrates decreased as the reaction temperature decreased. However, it was easy to see from Figure 6A,D,G that EK4-1 still maintained a high DH for the three substrates at lower temperatures (20 °C), retaining 80% of DH at 40 °C. The lower the temperature, the more obvious the advantage of EK4-1. Additionally, at 20 °C, the DH of the E423-digested substrate samples was only about 65% of that at 40 °C, while the DH of the EB62-treated sample only remained about 35% of that at 40 °C. These results indicated that with a decrease in temperature, the enzyme activity of protease E423 and EB62 decreased significantly, while protease EK4-1 could still maintain high proteolytic activity at a low temperature. These results indicated that the high catalytic efficiency of EK4-1 at a low temperature was superior to other homologous proteases and had good prospects for application.

A high degree of hydrolysis predicted that the protein substrates were digested into more oligopeptides and amino acids. More and more studies have shown that many hydrolysate oligopeptides possess a variety of biological activities, such as antioxidant, hypotensive, antibacterial and immune-promoting. Therefore, the hydrolysate products of grass fish skin and salmon skin were further studied for antioxidant analysis (Figure 6B,E). The results showed that both salmon skin and grass carp skin produced high levels of antioxidant peptides after proteolysis, and the hydrolysate product of protease EK4-1 had the highest antioxidant activity. The maximum DPPH and· OH scavenging activities of grass carp skin hydrolysis products via protease EK4-1 reached 68.05 ± 1.08% and 47.62 ± 1.52%, respectively, which were much higher than those of protease E423 and protease EB62. Similarly, the highest scavenging activity of DPPH and the ·OH of salmon skin hydrolysis products via protease EK4-1 was 47.38 ± 1.61% and 53.84 ± 3.01%, respectively.

When the substrate was grass carp skin, the peroxide radical scavenging activity in the products of the three proteases EK4-1, E423 and EB62 was 1.16 ± 0.05 mM·TE/g, 0.48 ± 0.03 mM·TE/g and 0.32 ± 0.03 mM·TE/g, respectively (Figure 6C).

While the substrate was the skin of salmon, the peroxide radical scavenging activity in the hydrolysis products of the three proteases was 0.79 ± 0.02 mM·TE/g, 0.59 ± 0.03 mM·TE/g and 0.36 ± 0.02 mM·TE/g, respectively (Figure 6F). When casein was the substrate, the hydrolyzed products of the three proteases had peroxyradical scavenging activity, and EK4-1 had the highest scavenging activity (0.31 ± 0.02 mM·TE/g), followed by EB62 (0.3 ± 0.01 mM·TE/g) and E423 (0.26 ± 0.01 mM·TE/g). These results suggested that the cold-adapted protease EK4-1 had important potential for the high-value modification of low-value protein resources at low temperatures.

## 3. Discussion

*Mesonia* belongs to the Flavobacteriaceae of Bacteroidetes, which was first proposed by Nedashkovskaya et al. in 2003. The discovery time was short, and the functions of a large number of genes have not been clarified [26]. At present, the situation of proteases in this genus has not been reported. This paper is the first systematic study of proteases in this genus, which is of great significance for people to understand and explore bacteria of the genus *Mesonia*. In this study, a novel M4 family metalloprotease EK4-1 with cold-adapted and salt-tolerant properties from an Arctic cryophilic bacterium *Mesonia algae* K4-1 was investigated. The protease EK4-1 had a low optimum proteolytic temperature, stable protein structure and high catalytic activity at a low temperature. It was found to be a typical cold-adapted protease. In addition, the high tolerance of the protease EK4-1 to NaCl, organic solvents, surfactants and oxidants has the potential to make it a bioenzyme additive compatible with detergent.

In the MEROPS database, the protease EK4-1 showed the highest sequence similarity with the identified Stearolysin from the M4 family of *Geobacillus stearothermophilus*, with a sequence identity of only 45%, indicating that the enzyme was relatively new in the amino acid sequence. EK4-1, as a metalloprotease with a typical HEXXH-conserved sequence, could specifically hydrolyze the substrate of the large hydrophobic and long-side-chain amino acids at the P1’ position, which is consistent with the characteristics of most M4 family proteases. Moreover, EK4-1 had more small-side-chain amino acids, fewer large-side-chain amino acids, more random coils and a larger activity pocket. These structural properties confirmed the flexibility of EK4-1, which might be an important reason for its high catalytic activity at low temperatures. In addition, the number of acidic amino acids in protease EK4-1 was far more than that of basic amino acids, which might be related to the salt tolerance of EK4-1.

By hydrolyzing a variety of low-value proteins at different temperatures, the superiority of cold-adapted protease EK4-1 for its high catalytic efficiency at low temperatures was demonstrated. In addition, the high antioxidant activity of these proteolytic products also provided a new reference for the application of cold-adapted protease EK4-1. This made it possible for the application of protease EK4-1 in the field of the high-value conversion of low-value proteins. In conclusion, the multi-tolerance and efficient proteolysis abilities at a low temperature of the cold-adapted protease EK4-1 indicate that EK4-1 has great potential for application in the field of low-temperature proteolysis.

## 4. Materials and Methods

### 4.1. Culture and Enzyme-Producing Fermentation of Strains

*Mesonia algae* K4-1 isolated from the Arctic Ocean (12°07.553′ E, 78°55.464′ N) was cultured in medium 2216E containing 0.5% (*m*/*v*) peptone (Oxoid, Basingstoke, UK), 0.1% (*m*/*v*) yeast extract (Oxoid, Basingstoke, UK) and 0.001% (*m*/*v*) Fe_2_(PO_4_)_3_ in 1 L artificial seawater, pH 7.5 [27]. When the strain grew to the logarithmic phase, 2% (*v*/*v*) of the bacteria culture solution was transferred to fermentation medium. The fermentation medium contained 2% bean powder, 2% corn powder, 1% wheat bran, 0.1% Na_2_CO_3_, 0.1% CaCl_2_, 0.1% Na_2_HPO_4_ and 0.03% KH_2_PO_4_ in artificial seawater of pH 8.0. The enzyme activity of fermentation broth was measured every 24 h after 120 h of continuous shaking of the culture. The samples were centrifuged at 12,000× *g* for 30 min, and the supernatant was stored at −20 °C for subsequent assay. The above reagents were purchased from Sinopharm Chemical ReagentCo., Ltd. (Shanghai, China), unless otherwise specified.

### 4.2. Protease Activity Assay and Protein Quantification

The protease activity of the culture supernatant on casein was detected using the Folin phenol method [10]. One unit of protease activity was determined as the amount of enzyme that catalyzed the formation of 1 µg tyrosine per min. The protein contained in the supernatant was quantified using the BCA protein quantification kit (Biosharp, Anhui, China), and bovine serum albumin (Biosharp, Anhui, China) at concentrations of 0, 0.2, 0.4, 0.6 and 1 mg/mL was used as the standard for the determination of the protein content [28].

### 4.3. Protease Purification and Mass Spectrum Identification

An appropriate amount of the fermentation broth was centrifuged at 12,000 rpm and the supernatant was collected. The 40% (*m*/*v*) ammonium sulfate powder was slowly added under an ice bath condition. The precipitate was collected via centrifugation, dissolved in 20 mM Tris-HCl (pH 8) and dialyzed. The samples were loaded onto a 5 mL HiTrapTM DEAE column (GE Healthcare) previously equilibrated with 20 mM Tris-HCl (pH 8.0) and then eluted with a NaCl gradient (0–1.0 M) in the same buffer at a flow rate of 1 mL/min. Active fractions were collected, concentrated via ultrafiltration (10 kDa MW cut-off membrane, Millipore) and then subjected to gel filtration on a Superdex 75 (10 × 300 column, 24 mL) with a flow rate 1 mL/min. The purity of protease EK4-1 was determined via SDS-PAGE as described previously. Substrate immersing zymography was modified to detect the proteolytic activities according to the method developed by Liu [29]. The intact target bands were cut into a centrifuge tube and sent to Sangon Biotech for NanoLC-ESI-MS/MS mass spectrometry identification. The identified sequences were aligned in the whole protein sequence encoded by Mesonia algae K4-1 and Espript (http://espript.ibcp.fr/ESPript/ESPript/ (accessed on 1 March 2023)) [19] was used for amino acid sequence alignment. The amino acid composition and domain of protease EK4-1 were analyzed using the ExPASy ProtParam (https://web.expasy.org/protparam/ (accessed on 1 March 2023)) tool [30] and InterPro database (https://www.ebi.ac.uk/interpro/ (accessed on 1 March 2023)) [31], respectively.

### 4.4. Characterization of Extracellular Protease EK4-1

#### 4.4.1. Optimum Temperature and Thermal Stability of Protease EK4-1

The protease was mixed with 2% casein in a ratio of 1:1 (*v*/*v*, mL/mL), incubated at different temperatures (4–60 °C) in 20 mM Tris-HCl buffer (pH 8.0) for 10 min. The optimal reaction temperature was determined by measuring the enzyme activity. The highest enzyme activity was set to 100%.

The thermal stability of the protease was determined by pre-incubating the protease in 20 mM Tris-HCl (pH 8.0) under 30–50 °C for 20 min, 40 min, 60 min, 90 min and 120 min, respectively [32]. The non-heated enzymes were taken as being 100% and the residual enzyme activity was measured at 40 °C in 20 mM Tris-HCl (pH 8.0) for 10 min with 2% (*w*/*v*, g/mL) casein.

The thermal properties of protease EK4-1 were analyzed using nano differential scanning calorimetry (DSC, TA Instruments, New Castle, DE, USA). Protease EK4-1 was diluted to approximately 2.5 mg/mL with 20mM Tris-HCl (pH 8.0). Experimental group: 500 µL of protease EK4-1 was added to the sample cell of DSC, and the same volume of 20 mM Tris-HCl (pH 8.0) was added to the reference cell. Blank group: both the reference and sample cells were filled with 500 µL of 20 mM Tris-HCl (pH 8.0). The heating rate was 1 °C/min from 20 °C to 90°C. Data were recorded automatically and subsequently analyzed using the NanoAnalyze Software provided by the manufacturer.

#### 4.4.2. Optimum pH, Acid–Base Stability and Salt Tolerance

The effect of pH on protease activity was studied by measuring its activity in the pH range of 4.0–11.0 using 2% (*w*/*v*, g/mL) casein as a substrate at 40 °C. The pH effect was investigated using one of the following buffers (0.02 M): citrate buffer (pH 4.0–7.0), Tris-HCl (pH 7.0–9.0) or glycine–NaOH (pH 9.0–11.0). The highest enzyme activity was set to 100%.

pH stability studies: The enzyme was mixed with pH buffers which included citrate buffer (pH 4.0–7.0), Tris-HCl (pH 7.0–9.0) and glycine–NaOH (pH 9.0–11.0) and was pre-incubated at 4 °C for 24 h. The residual activity was measured at 40 °C in 20 mM Tris-HCl (pH 8.0) with 2% (*w*/*v*, g/mL) casein for 10 min. The highest enzyme activity was set to 100%.

The effect of NaCl on the protease was investigated by measuring the protease activity at various NaCl concentrations (0–4.5 M) in 20 mM Tris-HCl buffer (pH 8.0). The protease activity of the purified enzyme was measured at 40 °C for 10 min with 2% (*w*/*v*, g/mL) casein. The relative activity was calculated by considering the enzyme activity without NaCl as being 100%.

Salt tolerance: Different concentrations of NaCl were mixed with EK4-1 in 20 mM Tris-HCl (pH 8.0) and stored at 4 °C for 48 h. The final concentration of NaCl was 0 M, 0.25 M, 0.5 M, 0.75 M, 1 M, 1.5 M, 2 M, 2.5 M, 3 M, 3.75 M, 4 M and 4.5 M, respectively. Then, the protease residual activity was measured at 40°C in 20 mM Tris-HCl (pH 8.0) with 2% (*w*/*v*, g/mL) casein for 10 min. The relative activity was calculated by considering the enzyme activity without NaCl as being 100%.

#### 4.4.3. Effect of Metal Ions, Oxidants, Surfactants, Organic Solvents and Inhibitor on Protease EK4-1 Activity

Solutions of K^+^ (KCl), Ca^2+^ (CaCl_2_), Mn^2+^ (MnSO_4_), Ba^2+^ (BaCl_2_), Fe^3+^ (FeCl_3_), Co^2+^ (CoCl_2_), Mg^2+^ (MgCl_2_), Zn^2+^ (ZnSO_4_), Cu^2+^ (CuSO_4_), Al^3+^ (AlCl_3_) and Fe^2+^ (FeCl_2_) in the quantity of 5 mM and 20 mM were prepared using Tris-HCl buffer (20 mM, pH 7.8). The diluted protease solution was mixed with various ionic solutions in a ratio of 1:1 (*v*/*v*, mL/mL), so that the final concentrations of various ions were 2.5 mM and 10.0 mM, respectively, and were stored at 4 °C for 24 h. The protease residual activity was measured at 40 °C in 20 mM Tris-HCl (pH 8.0) with 2% (*w*/*v*, g/mL) casein for 10 min. The relative activity was calculated by considering the enzyme activity without metal ions as being 100%.

Different concentrations of H_2_O_2_, Tween80, Triton X-100 and SDS were mixed with the same volume of protease at 4 °C for 30 min, respectively. The protease residual activity was measured at 40 °C with 2% (*w*/*v*, g/mL) casein for 10 min, and the untreated protease EK4-1 was used as a control (%).

The protease EK4-1 was preincubated with organic solvents ethanol, isopropanol, methanol, isoamyl alcohol, DMSO, xylene, propyl alcohol, n-hexane and n-butyl alcohol in a ratio of 1:3 (*v*/*v*, mL/mL) for 12 h at 4 °C. Then, the protease residual activity was measured at 40 °C with 2% (*w*/*v*, g/mL) casein for 10 min. The relative activities were calculated by considering the enzyme activity without organic reagents as being 100%.

Different concentrations of OP and protease EK4-1 were mixed in a ratio of 1:1 (*v*/*v*, mL: mL) to make the final concentrations of OP 0 mM, 2.5 mM and 10mM, respectively, and were stored at 4 °C for 24 h. Then, the protease residual activity was measured at 40 °C. The protease activity in the absence of an inhibitor was defined as being 100% (control). Inhibitor–substrate immersion enzyme zymography was based on Olena’s method, with slight modifications [33].

### 4.5. Structure Analysis

In order to investigate the structural specificity of the protease EK4-1, SOPMA (https://npsa-prabi.ibcp.fr/cgibin/npsa_automat.pl?page=/NPSA/npsa_sopma.html (accessed on 4 March 2023)) [34] was used to predict the secondary structure of the protease EK4-1 to obtain the amount of random coils. Three-dimensional structure molecular modeling of the protease EK4-1 was estimated using the program SWISS-MODEL (https://swissmodel.expasy.org/ (accessed on 4 March 2023)) [35]. Then, Proteins Plus (https://proteins.plus/ (accessed on 4 March 2023)) [36] was used to predict the active pocket of the cold-adapted protease EK4-1 and thermophilic protease Stearolysin to compare and analyze the structural and functional specificity of the cold-adapted protease EK4-1.

### 4.6. Enzyme Cleavage Site Analysis of Protease EK4-1

Insulin box substrate (5 mg/mL), protease solution (0.2 mg/mL) and 20 mM Tris-HCl (pH 8.0) were mixed at the ratio of 20:1:79 (*V*:*V*:*V*) and incubated for 1 h at 37 °C, and the hydrolysis products were analyzed using LC-MS/MS to determine the molecular weight of each peptide in the mixed peptide samples. The tool FindPept (https://web.expasy.org/findpept/ (accessed on 1 February 2023)) [37] was used to search the 30 amino acids of the insulin box and the molecular weight of the enzymatic peptides to obtain the possible amino acid composition of each peptide. Six dipeptide substrates (FA-Gly-Leu-NH_2_, FA-Gly-Phe-NH_2_, FA-Gly-Val-NH_2_, FA-Lys-Gly-OH, FA-Glu-Glu-OH and FA-Ala-Arg-OH) were synthesized by China Peptides Co., Ltd. (If these synthetic dipeptides are hydrolyzed via proteases, it leads to a decrease in the absorption value of 345 nm.

EK4-1 was mixed with the dipeptide substrate (1 mM) at a ratio of 1:1; then, the absorbance of the mixture was detected with a wavelength of 345 nm at 25 °C for 600 s. The absorbance values were monitored in real time using the time-course measurement software included with the UV-Vis Spectrophotometer Brochure (Cary 60, Agilent, USA).

Calculation of *k_cat_*/*K**_m_* values: The rate of decline of the absorbance value at 345 nm was used to indicate the speed of the enzymatic reaction.

According to *v* = *k_cat_*•[*E*]_0_•[*S*]_0_/*K_m_*, ([*S*]_0_<<*K_m_*_)_, the following formula was obtained:

*k_cat_/K_m_* = *v*/([*E*]_0_•[*S*]_0_) = *k*/(*b*•∆ε_345_)/([*E*]_0_•[*S*]_0_);

*k_cat_/K_m_*: apparent second-order-rate constant, s^−1^•M^−1^;

*v*: the reaction rate of the enzymatic reaction, M•s^−1^;

*k*: the absolute value of the decline rate of the absorbance value at 345 nm is the absolute value of the slope of the absorbance decline curve (the straight part), s^−1^;

*b*: cuvette thickness (a 1 cm thickness cuvette was used in this experiment);

∆ε_345_: molar extinction coefficient, −317 M^−1^cm^−1^;

[*E*]_0_: initial enzyme concentration of the reaction system, M;

[*S*]_0_: initial substrate concentration of the reaction system, M.

### 4.7. Degradation of Low-Value Protein via Cold-Adapted Protease EK4-1

Grass carp skin was sliced and sterilized at 121 °C for 20 min and then was digested with different proteases with same catalytic activity units (35 U/mL) at an enzyme to substrate ratio of 1:125 (*v*/*w*, mL/mg) and temperatures of 20 °C, 30 °C and 40 °C. The preparation method of salmon skin collagen was conducted according to Wu’s method [2]. Salmon skin was digested at an enzyme to substrate ratio of 1:10 (*v*/*w*, mL/mg). The 2% casein was prepared using Tris-HCl (20 mM, pH 8.0) and digested at an enzyme to substrate ratio of 1:10 (*v*/*w*, mL/mg). After the reaction for 10 h, the centrifuge tubes were transferred to a 90 °C water bath and inactivated for 10 min to stop the reaction. The hydrolysis degree at various treatment temperatures was analyzed using the ninhydrin coloration method. The standard curve was determined using leucine at concentrations of 0, 1, 1.5, 2.5 and 3 mM [2]. The DPPH free radical scavenging activity (DPPH RSA) assay, hydroxyl free radical scavenging activity (OH RSA) assay and oxygen radical absorbance capacity (ORAC) assay were conducted according to Liu’s method, with minor modifications [38]. The protease E423 used in this experiment was produced by *Pseudoalteromonas* sp. 423, and the protease EB62 was produced by *Pseudoalteromonas* sp. B62, whereby both of which are laboratory-preserved strains.

### 4.8. Statistical Analysis

The statistical analysis was performed using SPSS software (version 23, SPSS Inc. Chicago, IL, USA) and the data were presented as mean ± standard deviation (SD) of three replicates. The statistics between the two groups were analyzed using analysis of variance (ANOVA) and Duncan’s means comparison test using a significance level of *p* < 0.05. Charts were developed with the use of origin 2018 software (Northampton, MA, USA).

## Figures and Tables

**Figure 1 ijms-24-07985-f001:**
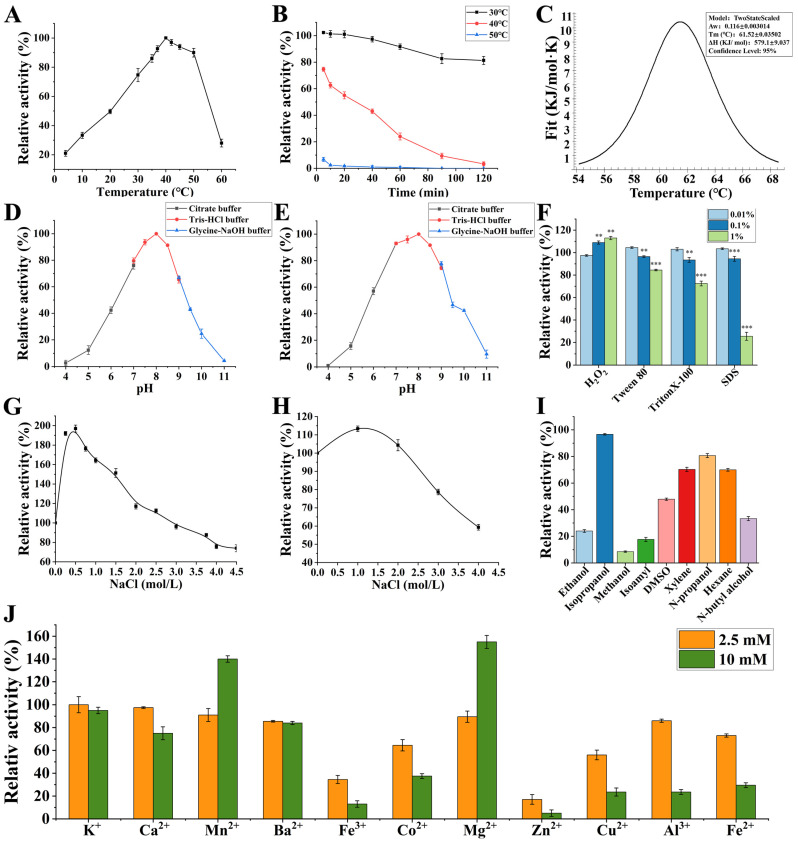
Enzymatic properties of EK4-1. (**A**) Optimal activity temperature: the temperature profile of protease EK4-1 activity was measured in 20 mM Tris-HCl buffer (pH 8.0) in the range of 4 °C to 60 °C for 10 min. The highest enzyme activity was set to 100%. (**B**) Thermal stability: protease EK4-1 was pre-incubated at temperatures ranging from 30 °C to 50 °C for 120 min, and then the residual activity of protease EK4-1 was determined in 20 mM Tris-HCl buffer (pH 8.0). The non-heated enzymes were taken as being 100%. (**C**) Tm value of EK4-1. (**D**) Optimal pH: the protease reaction was performed in 20 mM citrate buffer (pH 4.0–7.0), Tris-HCl buffer (pH 7.0–9.0) and glycine buffer (pH 9.0–11.0) at 40 °C for 10 min and determined using the folinphenol method. (**E**) pH stability: protease EK4-1 was diluted into buffers which were 20 mM citrate buffer (pH 4.0–7.0), Tris-HCl buffer (pH 7.0–9.0) and glycine buffer (pH 9.0–11.0) and stored at 4 °C for 24 h; then, the residual activity was measured. In (**D**,**E**), the highest enzyme activity was set to 100%. (**F**) Effects of oxidants and surfactants: protease EK4-1 was diluted into oxidants and surfactants at 4 °C for 30 min and then the residual activity was determined. The untreated enzyme was used as a control (100%). (**G**) Effects of NaCl concentration on EK4-1. The effect of NaCl on protease activity was measured under the same conditions in 20 mM Tris-HCl buffer (pH 8.0) in a range of 0–4.5 M NaCl. (**H**) Salt tolerance: protease EK4-1 was incubated at different concentrations of NaCl ranging from 0 M to 4.5 M at 4 °C for 48 h, and then the residual activity was determined. In (**G**,**H**), the relative activity was calculated by considering the enzyme activity without NaCl as being 100%. (**I**) Effects of organic reagents: protease EK4-1 was diluted into organic reagents at 4 °C for 30 min and then the residual activity was determined. The relative activities were calculated by considering the enzyme activity without organic reagents as being 100%. (**J**) Effects of metal ions: the effects of metal ions on enzyme activity were determined in 20 mM Tris-HCl buffer (pH 8.0) in the presence of each metal ion (final concentration of 2.5 mM and 10 mM) at 4 °C for 24 h, and then the residual activity was determined. The relative activity was calculated by considering the enzyme activity without metal ions as being 100%. All of the data shown above are the mean of at least three replicates (** *p* < 0.01 and *** *p* < 0.001).

**Figure 2 ijms-24-07985-f002:**
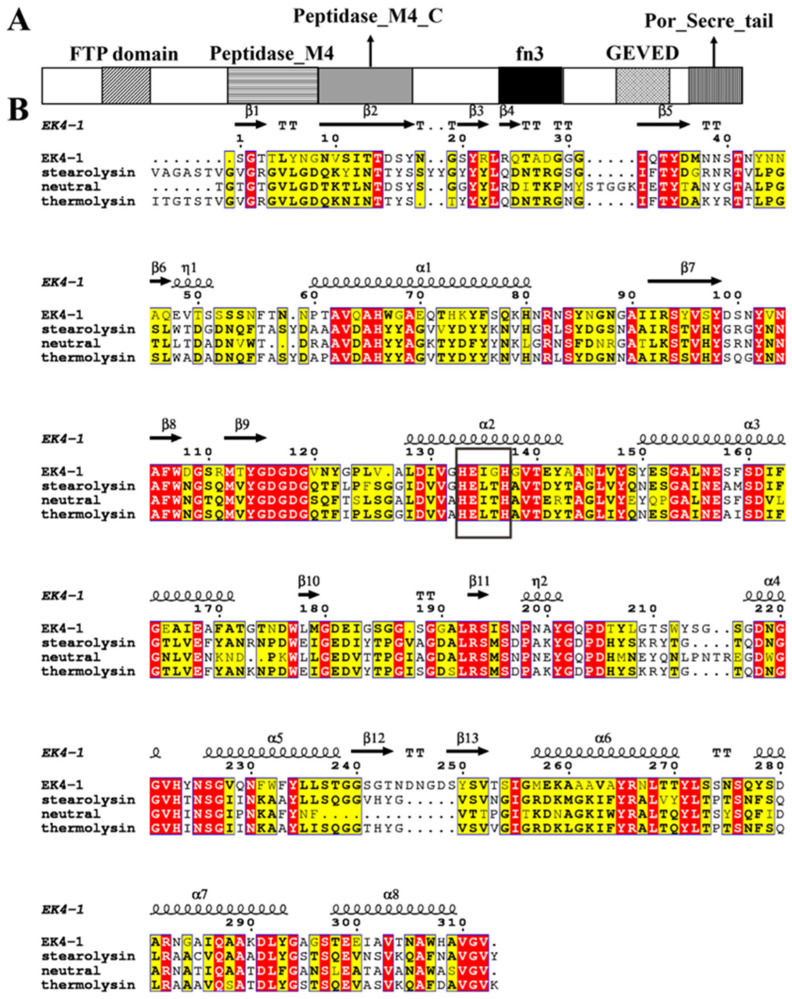
Sequence information of protease EK4-1. (**A**) The domain of the protease EK4-1. (**B**) Amino acid sequence homology analysis of EK4-1, Stearolysin, neutral peptidase and thermolysin (Stearolysin: identified from *Geobacillus stearothermophilus*; neutral peptidase: identified from *Thermoactinomyces* sp.27; thermolysin: identified from *Bacillus thermoproteolyticus*). Bold black box indicates conserved HEXXH motif on the second α-helix of the N-terminal domain. Red background indicates highly conserved sequence. Yellow background indicates similarity between groups.

**Figure 3 ijms-24-07985-f003:**
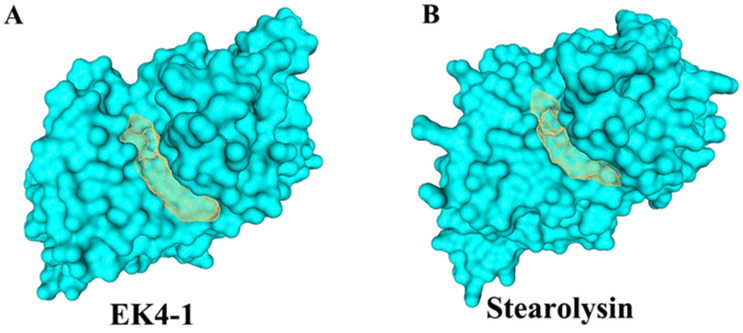
The 3D structure of protease EK4-1. (**A**) Substrate binding pocket for EK4-1. (**B**) Substrate binding pocket for Stearolysin.2.4. Analysis of the cleavage sites of protease EK4-1.

**Figure 4 ijms-24-07985-f004:**
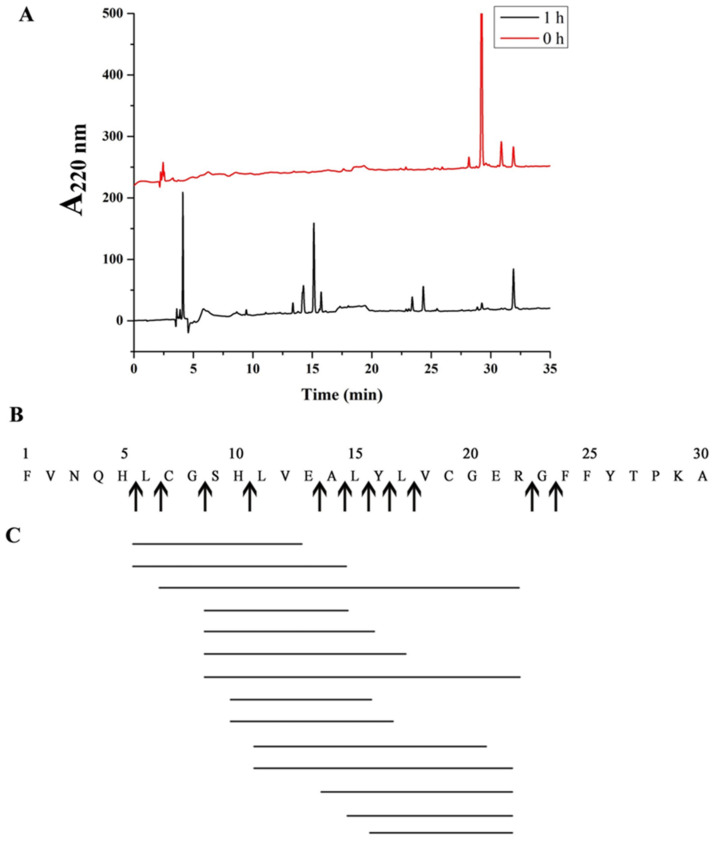
(**A**) Analysis of insulin box products via HPLC. Hydrolysis conditions were 37 °C, pH 8.0, Tris-HCl buffer. (**B**) The cleavage site of EK4-1 on the insulin box is represented by arrows. (**C**) Peptides produced via hydrolysis of insulin box using EK4-1.

**Figure 5 ijms-24-07985-f005:**
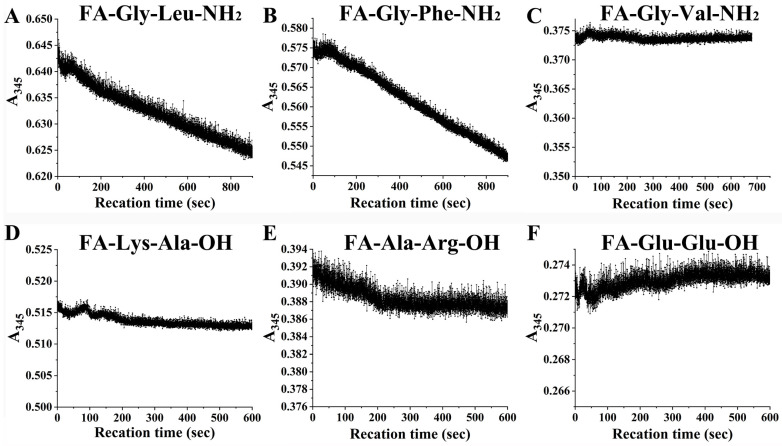
EK4-1 hydrolysis curves for six synthetic dipeptides. Substrates are (**A**) FA-Gly-Leu-NH_2_, (**B**) FA-Gly-Phe-NH_2_, (**C**) FA-Gly-Val-NH_2_, (**D**) FA-Lys-Ala-OH, (**E**) FA-Ala-Arg-OH and (**F**) FA-Glu-Glu-OH.

**Figure 6 ijms-24-07985-f006:**
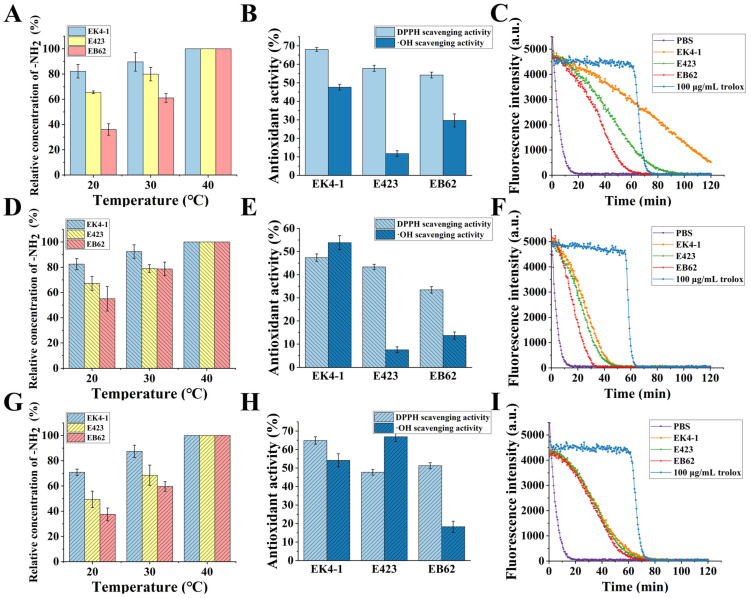
Analysis of the amount of -NH2 produced via enzymatic hydrolysis of the substrate and the antioxidation of the product. (**A**–**C**) The amount of -NH2, DPPH, hydroxyl radical scavenging activity and Peroxy radical scavenging activity; the substrate was grass carp skin. (**D**–**F**): The substrate was salmon skin and the other contents were the same as (**A**–**C**). (**G**–**I**): The substrate was casein and the other contents were the same as (**A**–**C**).

**Table 1 ijms-24-07985-t001:** Results of purification steps of protease EK4-1.

Purification Steps	Total Volume (mL)	Total Protein (mg)	Total Activity (U)	Specific Activity (U/mg)	Purification Fold	Yield (%)
Crude broth	1200	2062.5	129,846	62.95	1.00	100
Ammonium sulfate precipitation	120	117.25	82,044.58	699.74	11.12	63.19
Anion-exchange chromatography	100	50.36	42,721.21	848.24	13.47	32.90
Size-exclusion chromatography	50	10.94	9777.06	893.92	14.20	7.53

**Table 2 ijms-24-07985-t002:** Comparative analysis of special amino acid composition of protease.

Protease		EK4-1	E495	MCP-02	Stearolysin
Sources of strains		Arctic sea water	Arctic sea ice	Deep-sea sediment	Canned corn
Small side chain (%)	Gly	10.2	10.9	9.5	12.2
	Ser	12.5	12.1	9.9	5.8
	Thr	8.8	7.7	7.6	6.9
	Asn	8.4	8.9	10.3	5.5
	(Gly + Ser + Thr + Asn)	39.9	39.6	37.3	29.5
Large side chain (%)	Arg	1.6	1.8	2.1	5.1
	Leu	5.6	6.8	7.8	6.9
	Lys	4.3	3.8	4.0	3.3
	(Arg + Leu + Lys)	11.5	12.4	13.9	15.3
Acidic amino acid (%)	Asp	5.7	5.5	5.8	6.0
	Glu	4.8	3.6	3.4	4.2
	(Asp + Glu)	10.5	9.1	9.2	10.2
Basic amino acid (%)	Arg	1.6	1.8	2.1	5.1
	Lys	4.3	3.8	4.0	3.3
	His	1.5	1.2	1.2	2.2
	(Arg + Lys + His)	7.4	6.8	7.3	10.6
Acidic/basic amino acid	(Asp + Glu)/(Arg + Lys + His)	1.42	1.34	1.26	0.96
α-helix (%)		18.19	23.56	26.82	29.56
β-sheet (%)		29.78	22.60	23.66	22.08
Random coil (%)		48.07	47.12	43.47	42.88

**Table 3 ijms-24-07985-t003:** Comparative analysis of peptides generated by EK4-1 from insulin.

Peptide	Mr (Experimental) ^1^	Mr (Calculated) ^2^	Position
H.LCGSHLVE.A	904.4072	904.3960	6–13
H.LCGSHLVEAL.Y	1088.4974	1088.5172	6–15
L.CGSHLVEALYLVCGERG.F	1900.7811	1900.8295	7–23
G.SHLVEAL.Y	767.4086	767.4177	9–15
G.SHLVEALY.L	930.4758	930.4811	9–16
G.SHLVEALYL.V	1043.5594	1043.5651	9–17
G.SHLVEALYLVCGERG.F	1692.8301	1692.8141	9–23
S.HLVEALY.L	843.4443	843.4490	10–16
S.HLVEALYL.V	956.4919	956.5331	10–17
H.LVEALYLVCGER.G	1411.7626	1411.7017	11–22
H.LVEALYLVCGERG.F	1468.6918	1468.7232	11–23
E.ALYLVCGERG.F	1127.5095	1127.5281	14–23
A.LYLVCGERG.F	1056.5012	1056.4910	15–23
L.YLVCGERG.F	943.4132	943.4069	16–23

^1^ Mr (experimental) indicates the molecular mass of peptide after the charge is removed. ^2^ Mr (calculated) indicates the molecular mass of peptide calculated from the peptide sequence given by the Mascot Daemon. Position indicates the location of the peptide in the insulin box sequence.

**Table 4 ijms-24-07985-t004:** *k_cat_*/*K_m_* values of EK4-1 for synthetic substrates.

Synthetic Substrate	*k_cat_/K_m_* (mM^−1^S^−1^)	[*E*] (μM)	[*S*] (mM)
FA-Gly-Phe-NH_2_	3.513	0.0541	0.5
FA-Gly-Leu-NH_2_	2.925	0.0541	0.5

## Data Availability

The data presented in this study are available upon request from the corresponding author. The data are not publicly available due to privacy.

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
