# Peer review of "Characteristics and Application of a Novel Cold-Adapted and Salt-Tolerant Protease EK4-1 Produced by an Arctic Bacterium Mesonia algae K4-1"

_ijms, 2023, doi:10.3390/ijms24097985_

Round 1

Reviewer 1 Report

In the manuscript submitted to me for review entitled: „Characteristics and application of a novel cold-adapted and salt-tolerant protease EK4-1 produced by Arctic bacteria Mesonia algae K4-1 the authors present the properties of a cold-adapted protease EK4-1 secreted by Mesonia algae K4-1 from Arctic. The proven properties of EK4-1 have broad prospects for application in the cosmetic and detergent industries.

My remarks and recommendations to the authors are:

1.     To correct the numbering of the authors' affiliations at the beginning in the order of introduction of the authors.

    They should be:

2) Key Laboratory of Functional Dairy;

5) Zhong Shi Du Qing (Shandong) Biotechnology Company

2.     The legends of figures 1B, 1C, 1D, 1E and 1F are too small and illegible. Let them be enlarged and easily visible to the reader.

3.     In the description of figure 1 at the end, the meanings of the "p" values are indicated, but I did not see where in the figure itself "p" is presented as a statistic.

4.     In figure 2, do figure A and figure B have to be part of one common figure? The inscriptions on the left side of Figure 2A thus presented are not readable. In my opinion, if 2A and 2B are given as separate figures and 2A is enlarged, the inscriptions should be seen better.

5.     The legends in figure 5 everywhere are illegibly small - let them increase.

6.      In section 4.1 it is not indicated where medium 2216E, peptone and yeast extract were purchased or provided, as well as in section 4.2. the source of bovine serum albumin (company, city, country) is not indicated.

Reviewer 2 Report

This paper describes the characterization of a cold-adapted and salt-tolerant protease EK4-1 of an Arctic bacterium, Mesonia algae K4-1. This paper contains interesting contents in its own way, but also contains some points to be corrected, supplemented, or revised. My main concerns are as described below.

1.       Overall, the resolution of the figures is too low. Figure 2, 4, etc. are too low resolution to read

2.       Overall, this paper has a problem in describing the results obtained from real experiments and the predictions through algorithms without clear distinction. What is predicted by the algorithm must be described as “~estimated” or “~predicted”.

3.       In addition, the description of how the experiment was conducted is vague and lacking. For example, in the case of an experiment in which metal ions were added and enzyme activity was measured (Figure 1J), the Materials and Methods only indicated “20 mM solutions of K+, Ca2+, Mn2+, Ba2+, Fe3+, Co2+, Mg2+, Zn2+, Cu2+, Al3+ and Fe2+ were prepared with Tris-HCl buffer (20 mM, pH 7.8).” This means that the final state of the solution is like this, not a description of how it was prepared. Authors should state exactly what and how much you put in so that others can reproduce this experiment. In this way, it is possible for readers to figure out what and how much anion corresponds to the cation in the solution used in the actual experiment.

4.       In D and E of Fig. 2, the x and y axes are the same. So, two graphs represent the same thing. However, according to figure legend, D presents enzyme activity, and E presents enzyme stability. There should be a clear explanation how differently each experiment was carried out, and a description of the difference, either in the graph itself or in the legend. The same problem occurs in G and H as well.

5.       185-205. Although this paper is discussing a lot about protein flexibility, how to obtain protein flexibility is not described in Materials and Methods or figure legend.

6.       213-220. It is wrong to describe it as if it was a result obtained through an experiment, saying “The experimental results showed~” even though it is said that it was calculated on the Swiss-Model website.

7.       In Discussion, the authors also said “the protease EK4-1 showed the highest sequence similarity with the identified Stearolysin from the M4 family of Geobacillus stearothermophilus, with a sequence identity of only 45%, indicating that the enzyme was relatively new in amino acid sequence.” As it is said, at least one experimental confirmation is needed rather than claiming that this protein is a metaloprotease based on amino acid homology alone. Usually, treatment with chelating agents such as EDTA leads to complete inactivation of the metaloproteases. For example, EDTA is a metal chelator that removes zinc, which is essential for activity. They are also inhibited by the chelator orthophenanthroline. Experiments with these agents are needed for confirmation. The authors' experiment in Figure 1J does not show this clearly because it is an experiment with metal ions added, not removed.

8.       What is a low-value protein? Authors should indicate which of the grass fish skin, salmon skin, and casein used is (are) a low-value protein and which is (are) a high-value protein.

9.       305, “Similar results were found in the hydrolysis products of casein.” It is ambiguous whether the similar result in this sentence is similar to the case of DPPH or hydroxy radical, or similar to E423 and EB62. I think it means something similar to E423 and EB62, but it should be clarified. In addition, the subtitle of this part “Efficiently degrade low-value protein substrates by EK4-1 at low temperature” or the final remark of this part “These results suggested that the cold-adapted protease EK4-1 had important potential for high-value modification of low-value protein resources at low temperatures” to be valid, the difference in results between low-value and high-value proteins must be clearly stated.

10.   In Fig. 1A, the activity of EK4-1 at 40°C is higher than at 30°C. Why is the activity at 30°C higher in Fig. 1B?

11.   261-262. “Previous studies results had shown that the cold-adapted protease EK4-1 has high catalytic efficiency at low temperature and multiple cleavage site.” If this is about the previous results of this paper, it should be called previous results, not previous studies. If it is really the contents of a previous study that has already been published in previous papers, the paper should be cited as a reference.

Minor concerns are:

1.       In titile, “Arctic bacteria” should be amended to “an Arctic bacterium”

2.       The first sentence of the abstract seems redundant

3.       44. No period at the end of a sentence

4.       124. double space between sentences

Round 2

Reviewer 2 Report

This paper describes the characterization of a cold-adapted and salt-tolerant protease EK4-1 of an Arctic bacterium, Mesonia algae K4-1. Various supplements were requested for the original paper, and it seems that these supplements were generally well done in the revised paper. Especially, the experimental methods and presentations that were somewhat insufficient in the original manuscript seem to have been supplemented well. However, in their revised paper, Fig. 6 just seem to be removed. I think new Fig. 6 should be presented.
